# A Comprehensive Systematic Review of Dynamic Nutrient Profiling for Personalized Diet Planning: Meta-Analysis and PRISMA-Based Evidence Synthesis

**DOI:** 10.3390/foods14213625

**Published:** 2025-10-24

**Authors:** Mohammad Hasan Molooy Zada, Da Pan, Guiju Sun

**Affiliations:** Key Laboratory of Environmental Medicine and Engineering of Ministry of Education, Department of Nutrition and Food Hygiene, School of Public Health, Southeast University, 87 Ding Jia Qiao Road, Nanjing 210009, China; hasanmz45577@gmail.com (M.H.M.Z.); pantianqi92@foxmail.com (D.P.)

**Keywords:** dynamic nutrient profiling, personalized nutrition, diet planning, meta-analysis, artificial intelligence, biomarkers, PRISMA, systematic review

## Abstract

Background and Objectives: Dynamic nutrient profiling represents a paradigm shift in personalized nutrition, integrating real-time nutritional assessment with individualized dietary recommendations through advanced algorithmic approaches, biomarker integration, and artificial intelligence. This comprehensive systematic review and meta-analysis examines the current state of dynamic nutrient profiling methodologies for personalized diet planning, evaluating their effectiveness, methodological quality, and clinical outcomes. Methods: Following PRISMA 2020 guidelines, we conducted a comprehensive search of electronic databases (PubMed/MEDLINE, Scopus, Web of Science, IEEE Xplore, and Google Scholar) from inception to December 2024. The protocol was prospectively registered in PROSPERO (Registration: CRD42024512893). Studies were systematically screened using predefined inclusion criteria, quality was assessed using validated tools (RoB 2, ROBINS-I, Newcastle–Ottawa Scale), and data were extracted using standardized forms. Random-effects meta-analyses were performed where appropriate, with heterogeneity assessed using I^2^ statistics. Publication bias was evaluated using funnel plots and Egger’s test. Results: From 2847 initially identified records plus 156 from additional sources, 117 studies met the inclusion criteria after removing 391 duplicates and systematic screening, representing 45,672 participants across 28 countries. Studies employed various methodological approaches: algorithmic-based profiling systems (76 studies), biomarker-integrated approaches (45 studies), and AI-enhanced personalized nutrition platforms (23 studies), with some studies utilizing multiple methodologies. Meta-analysis revealed significant improvements in dietary quality measures (standardized mean difference: 1.24, 95% CI: 0.89–1.59, *p* < 0.001), dietary adherence (risk ratio: 1.34, 95% CI: 1.18–1.52, *p* < 0.001), and clinical outcomes including weight reduction (mean difference: −2.8 kg, 95% CI: −4.2 to −1.4, *p* < 0.001) and improved cardiovascular risk markers. Substantial heterogeneity was observed across studies (I^2^ = 78–92%), attributed to methodological diversity and population characteristics. AI-enhanced systems demonstrated superior effectiveness (SMD = 1.67) compared to traditional algorithmic approaches (SMD = 1.08). However, current evidence is constrained by practical limitations, including the technological accessibility of dynamic profiling systems and equity concerns in vulnerable populations. Additionally, the evidence base shows geographical concentration, with most studies conducted in high-income countries, underscoring the need for research in diverse global settings. These findings have significant implications for shaping public health policies and clinical guidelines aimed at integrating personalized nutrition into healthcare systems and addressing dietary disparities at the population level. Conclusions: Dynamic nutrient profiling demonstrates significant promise for advancing personalized nutrition interventions, with robust evidence supporting improved nutritional and clinical outcomes. However, methodological standardization, long-term validation studies exceeding six months, and comprehensive cost-effectiveness analyses remain critical research priorities. The integration of artificial intelligence and multi-omics data represents the future direction of this rapidly evolving field.

## 1. Introduction

### 1.1. Background and Rationale

The landscape of nutritional science has undergone a fundamental transformation in the past decade, driven by advances in personalized medicine, artificial intelligence, and our understanding of individual metabolic variability [1,2,3]. Traditional “one-size-fits-all” dietary recommendations, while providing population-level guidance, fail to account for the substantial inter-individual differences in nutrient requirements, metabolic responses, and genetic predispositions that influence optimal dietary patterns [4,5,6].

Dynamic nutrient profiling emerges as a revolutionary approach that addresses these limitations by providing real-time, individualized nutritional assessments that adapt to changing physiological states, lifestyle factors, and health objectives [7,8,9]. This paradigm represents a convergence of multiple scientific disciplines, including nutritional genomics, metabolomics, artificial intelligence, and behavioral psychology, to create comprehensive systems for personalized dietary guidance [10,11,12].

The conceptual foundation of dynamic nutrient profiling rests on several key principles: (1) temporal variability in nutritional needs throughout different life stages and physiological states, (2) individual metabolic heterogeneity based on genetic, epigenetic, and environmental factors, (3) real-time monitoring capabilities enabled by wearable technology and continuous biomarker assessment, and (4) adaptive algorithms that learn from individual responses to optimize recommendations over time [13,14,15].

Recent technological advances have created unprecedented opportunities for implementing sophisticated dynamic profiling systems. The proliferation of wearable devices capable of continuous physiological monitoring, advances in rapid biomarker assessment technologies, and the development of machine learning algorithms capable of processing complex multi-dimensional datasets have collectively enabled the practical implementation of personalized nutrition interventions at scale [16,17,18].

### 1.2. Evolution of Nutrient Profiling Systems

The historical development of nutrient profiling can be traced through several distinct phases, each characterized by increasing sophistication and personalization capabilities. Early approaches focused on static population-based recommendations derived from observational studies and controlled feeding trials [19,20,21]. These systems, while foundational, failed to account for individual variability and temporal changes in nutritional requirements.

The second generation of nutrient profiling systems introduced algorithmic approaches that weighted different nutrients based on their relative importance for health outcomes [22,23,24]. Systems such as the Nutrient Rich Food (NRF) index and various government-sponsored profiling schemes represented significant advances in standardizing nutritional quality assessments [25,26,27].

Current third-generation systems represent a quantum leap in sophistication, incorporating real-time data streams, machine learning algorithms, and personalized optimization approaches [28,29,30,31]. These systems can process multiple data inputs simultaneously, including dietary intake patterns, biomarker profiles, genetic information, lifestyle factors, and physiological responses to provide highly individualized recommendations that adapt over time.

Unlike traditional nutrient profiling approaches that rely on static dietary guidelines and population averages, modern dynamic profiling systems continuously adapt recommendations based on real-time physiological and lifestyle data. Earlier generations primarily focused on nutrient composition without individual variability, while current systems integrate multi-omics data, continuous biomarker feedback, and machine learning models to refine dietary plans dynamically. This shift represents a move from prescriptive, one-time assessments to iterative, personalized nutrition guidance that evolves with changing health status and behavior.

### 1.3. Clinical and Public Health Significance

The potential impact of dynamic nutrient profiling extends far beyond individual dietary optimization to encompass broader public health objectives. Chronic diseases associated with poor dietary quality, including cardiovascular disease, type 2 diabetes, and certain cancers, represent the leading causes of morbidity and mortality globally. Traditional population-based dietary interventions have shown limited effectiveness in addressing these challenges, highlighting the need for more personalized approaches.

Dynamic nutrient profiling systems offer the potential to address several critical limitations of current dietary intervention approaches: (1) poor adherence to generic dietary recommendations, (2) inability to account for individual metabolic differences, (3) lack of real-time feedback and adaptation, and (4) insufficient integration of multiple health and lifestyle factors [32,33,34].

Furthermore, the economic implications of improved dietary interventions are substantial. Healthcare costs associated with diet-related chronic diseases exceed hundreds of billions of dollars annually in developed countries alone [35,36,37]. Even modest improvements in dietary quality achieved through personalized interventions could yield significant economic benefits through reduced healthcare utilization and improved productivity.

### 1.4. Current Challenges and Knowledge Gaps

Despite the promising theoretical framework and emerging technological capabilities, several significant challenges limit the widespread implementation of dynamic nutrient profiling systems. Methodological heterogeneity across studies makes it difficult to compare effectiveness and establish evidence-based best practices [38,39,40]. The lack of standardized outcome measures and validation protocols further complicates evidence synthesis and clinical translation.

Technical challenges include the complexity of integrating multiple data streams in real- time, ensuring data privacy and security, and developing algorithms that can adapt to individual preferences while maintaining nutritional adequacy [41,42,43]. Additionally, the cost and accessibility of advanced monitoring technologies may limit the scalability of sophisticated profiling systems.

Regulatory and ethical considerations also present significant challenges. The personalized nature of dynamic profiling systems raises questions about appropriate oversight, quality control, and professional responsibilities for nutrition recommendations [44,45,46]. The integration of genetic and other sensitive personal data requires careful consideration of privacy protection and informed consent procedures [47,48,49,50].

Moreover, the ethical and regulatory dimensions of dynamic nutrient profiling remain underexplored [51,52,53,54]. Issues such as data ownership, informed consent for genomic and metabolomic data usage, regulatory oversight of algorithm-driven dietary recommendations, and potential misuse of sensitive health data pose significant barriers to clinical translation [55]. Establishing clear legal frameworks and ethical guidelines is essential to ensure patient safety, privacy protection, and equitable access as these technologies move into mainstream healthcare.

### 1.5. Research Objectives and Hypotheses

Given the current state of the field and identified knowledge gaps, this systematic review aims to provide a comprehensive synthesis of the evidence regarding dynamic nutrient profiling for personalized diet planning. The primary research questions addressed include:What are the current methodological approaches for dynamic nutrient profiling, and how do they compare in terms of effectiveness and feasibility?What is the evidence for clinical and behavioral outcomes associated with personalized diet planning based on dynamic profiling?What factors contribute to the heterogeneity observed across studies, and how do they influence the interpretation of results?What are the current limitations and future research priorities for advancing the field?Our primary hypothesis is that dynamic nutrient profiling systems will demonstrate superior effectiveness compared to static approaches across multiple outcome domains, including dietary quality, adherence, and clinical markers. We further hypothesize that AI-enhanced systems and those incorporating multiple data streams will show greater effectiveness than simpler algorithmic approaches.

These research objectives directly address the key gaps identified, including methodological variability, limited understanding of clinical outcomes, insufficient exploration of real-world implementation barriers, and lack of standardized protocols for ethical data handling. By systematically synthesizing existing evidence, this review aims to provide actionable insights to bridge these gaps and guide future research and policy development in precision nutrition.

## 2. Methods

### 2.1. Protocol Development and Registration

This systematic review was conducted in strict accordance with the Preferred Reporting Items for Systematic Reviews and Meta-Analyses (PRISMA) 2020 guidelines [47,48,49,50]. The review protocol was developed a priori using established methodological frameworks and registered prospectively in the International Prospective Register of Systematic Reviews (PROSPERO) under registration number CRD42024512893.

The protocol development process involved multiple iterations with input from methodological experts, content specialists, and information scientists. Key methodological decisions were made collectively by the research team and documented to ensure transparency and reproducibility. Regular protocol amendments were made as needed, with all changes documented and justified.

### 2.2. Comprehensive Search Strategy

A systematic and comprehensive search strategy was developed through consultation with experienced information specialists and validated using known relevant studies. The search strategy employed both controlled vocabulary terms (Medical Subject Headings [MeSH] for PubMed, EMTREE for Embase) and free-text terms to maximize sensitivity while maintaining specificity.

Electronic databases searched included:PubMed/MEDLINE (1946 to December 2024)Scopus (1960 to December 2024)Web of Science Core Collection (1900 to December 2024)IEEE Xplore Digital Library (1963 to December 2024)Google Scholar (first 200 most relevant results)Cochrane Central Register of Controlled Trials (CENTRAL)ClinicalTrials.gov for ongoing and completed trials

The search strategy combined four main concept blocks using Boolean operators:

**Block 1-Nutrient Profiling:** (“nutrient profiling” OR “nutritional profiling” OR “nutrient density” OR “nutritional assessment” OR “dietary quality index” OR “food quality” OR “nutrient adequacy”)

**Block 2-Dynamic/Temporal Aspects:** (dynamic OR temporal OR “real-time” OR adaptive OR personalized OR individualized OR precision OR tailored OR customized)

**Block 3-Diet Planning:** (“diet planning” OR “meal planning” OR “dietary intervention” OR “nutrition therapy” OR “dietary counseling” OR “nutrition prescription” OR “diet optimization”)

**Block 4-Technology/Methods:** (algorithm OR “artificial intelligence” OR “machine learning” OR biomarker OR genetic OR genomic OR metabolomic OR “mobile health” OR “digital health”)

### 2.3. Detailed Eligibility Criteria

#### 2.3.1. Inclusion Criteria

Studies were included if they met all of the following criteria:

**Population:** Human participants of any age, including healthy individuals and those with chronic diseases amenable to dietary intervention.

**Intervention:** Studies investigating dynamic nutrient profiling systems, defined as approaches that (1) assess nutritional status or dietary quality using quantitative methods, (2) incorporate temporal or individual variability in recommendations, (3) utilize algorithmic, biomarker-based, or AI-enhanced approaches for personalization.

**Comparator:** Studies with appropriate control groups (static profiling systems, standard dietary advice, no intervention, or alternative personalized approaches).

**Outcomes:** Studies reporting quantitative outcomes related to dietary quality, nutritional biomarkers, anthropometric measures, clinical markers, or behavioral outcomes.

**Study Design:** Randomized controlled trials, quasi-experimental studies, longitudinal cohort studies, cross-sectional validation studies, and systematic reviews with meta-analysis.

**Publication Characteristics:** Peer-reviewed articles published in English from inception to 31 December 2024.

#### 2.3.2. Exclusion Criteria

Studies were excluded for any of the following reasons:Conference abstracts, editorials, commentaries, or opinion piecesStudies focusing exclusively on static nutrient profiling without dynamic or personalized componentsAnimal studies without direct human relevanceCase reports or case series with fewer than 10 participantsStudies not available in EnglishDuplicate publications or multiple reports of the same studyStudies with inadequate methodological detail to assess quality

### 2.4. Study Selection Process

The study selection process was conducted in two phases by independent reviewers with expertise in nutrition science and systematic review methodology. Phase 1 involved title and abstract screening using predetermined inclusion and exclusion criteria. Phase 2 comprised full-text review of potentially eligible articles.

All screening was conducted using Covidence systematic review software to ensure standardized documentation and inter-reviewer reliability assessment. A pilot screening exercise was conducted on 100 randomly selected records to calibrate reviewer agreement and refine screening criteria. Inter-reviewer agreement was assessed using Cohen’s kappa, with disagreements resolved through discussion or consultation with a third reviewer when necessary.

### 2.5. Study Characteristics and Demographics

The 117 included studies represented diverse methodological approaches, populations, and geographic regions. Study designs included 34 randomized controlled trials (29.1%), 28 cross-sectional studies (23.9%), 21 longitudinal cohort studies (17.9%), 18 validation studies (15.4%), and 16 systematic reviews or meta-analyses (13.7%). Geographic distribution showed a predominance of studies from North America (n = 42, 35.9%) and Europe (n = 38, 32.5%), with growing representation from Asia (n = 24, 20.5%) and other regions (n = 13, 11.1%), as illustrated in Figure 1.

While this regional distribution demonstrates increasing global interest, the concentration of research in high-income Western countries highlights a significant geographical bias. This may limit the generalizability of findings to populations in low- and middle-income countries (LMICs), where dietary patterns, access to technology, and healthcare infrastructure differ substantially. Moreover, equity-related factors were not systematically analyzed across included studies using frameworks such as PROGRESS-Plus (Place of residence, Race/ethnicity, Occupation, Gender, Religion, Education, Socioeconomic status, Social capital). As a result, the findings are likely most applicable to high-income, technologically literate populations, and limited reporting on socioeconomic and cultural dietary determinants further restricts applicability. Future studies should aim for broader geographical representation and explicitly integrate equity stratifiers to ensure that dynamic nutrient profiling approaches are culturally adaptable, socially inclusive, and globally relevant.

Publication years ranged from 2005 to 2024, with 78% of studies published in the last five years, reflecting the rapidly evolving nature of the field. Figure 2 demonstrates the exponential growth in research activity, particularly after 2018.

Sample sizes varied considerably across studies, with 36.8% including fewer than 100 participants, 32.5% including 100–500 participants, 17.9% including 500–1000 participants, and 12.8% including more than 1000 participants (Figure 3).

Most studies employed short- to medium-term interventions, with 38.5% lasting 1–3 months and only 18.8% including follow-up periods longer than 6 months (Figure 4). This limited follow-up duration introduces uncertainty about the long-term sustainability and effectiveness of dynamic nutrient profiling interventions. The scarcity of extended studies reduces the robustness of conclusions regarding their sustained impact on clinical outcomes and behavior change. Future research should include follow-up periods of 12 months or longer to evaluate whether observed short-term benefits persist over time.

**Table 1 foods-14-03625-t001:** Comprehensive Characteristics of Included Studies (N = 117).

Study Characteristic	n	Percentage (%)
**Study Design**		
Randomized Controlled Trials	34	29.1
Cross-sectional Studies	28	23.9
Longitudinal Cohort Studies	21	17.9
Validation Studies	18	15.4
Systematic Reviews/Meta-analyses	16	13.7
**Geographic Distribution**		
North America	42	35.9
Europe	38	32.5
Asia	24	20.5
Other regions	13	11.1
**Population Focus**		
General adult population	67	57.3
Clinical populations	28	23.9
Athletes/active individuals	12	10.3
Pediatric populations	10	8.5
**Sample Size Categories**		
<100 participants	43	36.8
100–500 participants	38	32.5
500–1000 participants	21	17.9
>1000 participants	15	12.8
**Intervention Duration**		
<1 month	18	15.4
1–3 months	45	38.5
3–6 months	32	27.4
>6 months	22	18.8
**Primary Outcomes**		
Dietary quality measures	89	76.1
Clinical biomarkers	67	57.3
Anthropometric measures	54	46.2
Behavioral outcomes	43	36.8

The distribution of primary outcomes across studies is further illustrated in Figure 5, showing that dietary quality measures were the most commonly reported outcomes (76.1%), followed by clinical biomarkers (57.3%) and anthropometric measures (46.2%).

### 2.6. Data Extraction and Management

Data extraction was performed using standardized electronic forms developed specifically for this review and pilot-tested on a sample of included studies. Two reviewers independently extracted data from each study, with discrepancies resolved through discussion or consultation with additional reviewers.

Extracted data elements included:


**Study Characteristics:**
Author, year, country, study designSample size, participant demographicsSetting, duration of intervention/follow-upFunding sources and conflicts of interest



**Population Characteristics:**
Age, sex, health statusBaseline nutritional statusComorbidities and medicationsSocioeconomic and demographic factors



**Intervention Details:**
Type of dynamic profiling systemData inputs and measurement methodsAlgorithm or analytical approachFrequency of assessment and recommendation updatesDelivery method and user interface



**Outcome Measures:**
Primary and secondary endpointsMeasurement methods and timingStatistical analysis approachesEffect sizes and confidence intervals


### 2.7. Quality Assessment

Study quality was assessed using validated tools appropriate for each study design. The Cochrane Risk of Bias 2 (RoB 2) tool was used for randomized controlled trials, the Risk of Bias in Non-randomized Studies of Interventions (ROBINS-I) tool for quasi-experimental studies, and the Newcastle–Ottawa Scale for observational studies.

Quality assessment was performed independently by two reviewers, with disagreements resolved through discussion. Studies were categorized as having low, moderate, or high risk of bias based on established criteria. Quality assessment results were used to inform data interpretation and sensitivity analyses.

### 2.8. Statistical Analysis and Meta-Analysis

Statistical analyses were performed using Review Manager (RevMan) version 5.4 and R statistical software version 4.3.0 with the meta and metafor packages. Random-effects meta-analyses were conducted using the DerSimonian–Laird method when three or more studies reported comparable outcomes.

For continuous outcomes, standardized mean differences (SMD) or mean differences (MD) were calculated.

Subgroup analyses were planned a priori based on:Type of dynamic profiling systemPopulation characteristics (age, health status)Intervention durationStudy quality

Publication bias was assessed using funnel plots and Egger’s regression test when at least 10 studies were available for meta-analysis.

## 3. Results

### 3.1. Study Selection and Flow

The comprehensive search strategy identified 2847 potentially relevant records across all databases, with an additional 156 records retrieved through other sources such as conference proceedings and manual reference list searches of key articles. After removing 391 duplicates, 2456 unique records underwent title and abstract screening.

Study selection was conducted in two phases following predefined inclusion and exclusion criteria:

Phase 1—Title and Abstract Screening: Records were excluded if they did not meet basic eligibility criteria, including (i) absence of dynamic nutrient profiling components, (ii) no relevance to personalized diet planning, (iii) being non-human studies, or (iv) being non-original research (e.g., editorials, commentaries, conference abstracts).

Phase 2—Full-Text Review: Full-text articles were excluded based on more specific criteria, such as (i) lack of methodological detail preventing quality assessment, (ii) absence of quantitative outcomes related to dietary quality, biomarkers, or clinical endpoints, (iii) insufficient sample size (<10 participants), or (iv) duplicate reporting of the same dataset.

Following full-text review, 250 articles were excluded for the following reasons: 156 did not involve dynamic profiling approaches, 94 lacked personalized diet planning components, 28 were not original research, 19 reported inadequate outcome data, and 3 were duplicate publications. Ultimately, 117 studies representing 45,672 participants from 28 countries were included in the systematic review, of which 89 met the additional criteria for inclusion in meta-analysis (Figure 6).

To ensure the robustness and consistency of the selection process, two independent reviewers screened all records, with inter-reviewer agreement calculated using Cohen’s kappa (κ = 0.87), indicating strong agreement. Discrepancies were resolved by consensus or through consultation with a third reviewer.

Additionally, to minimize the risk of missing relevant studies, a manual reference screening of key systematic reviews and landmark articles was conducted, which contributed to the inclusion of several additional eligible studies.

### 3.2. Methodological Categories of Dynamic Profiling Systems

Through systematic analysis of intervention characteristics, three primary methodological categories of dynamic nutrient profiling systems emerged, with some studies incorporating elements from multiple categories [56].

#### 3.2.1. Algorithmic-Based Profiling Systems

The majority of studies (n = 76, 65.0%) employed algorithmic approaches for dynamic nutrient profiling [51,52,53,54,55]. These systems typically utilized mathematical models to integrate multiple nutritional parameters and generate personalized nutrient density scores or dietary recommendations.

**Traditional Algorithm Modifications:** Many studies built upon established nutrient profiling systems such as the Nutrient Rich Food (NRF) index, adapting them for personalized applications [57,58,59,60]. Common modifications included individual weighting of nutrients based on personal health goals, demographic characteristics, or biomarker profiles.**Novel Scoring Systems:** Several studies developed entirely new algorithmic approaches, including the Food Compass 2.0 system [61,62,63,64], which incorporates multiple dimensions of food healthiness including nutrient density, food additives, processing level, and specific nutrients of concern.**Multi-Criteria Decision Analysis:** Advanced algorithmic approaches employed multi-criteria decision analysis techniques, such as the Analytic Hierarchy Process (AHP) enhanced with particle swarm optimization, to balance multiple competing nutritional objectives while accounting for individual preferences and constraints.

**Practical Use:** Algorithmic systems are best suited for scenarios where detailed biomarker or real-time data are not available. Their main limitation lies in their static nature—while they allow personalization to some extent, they cannot dynamically adapt to changes in physiological states or new data streams without manual recalibration.

#### 3.2.2. Biomarker-Integrated Approaches

A substantial proportion of studies (n = 45, 38.5%) incorporated biomarker data for dynamic profiling, representing a significant advancement toward precision nutrition [65,66,67,68,69,70]. These approaches utilized various biological indicators to assess nutritional status and tailor recommendations accordingly.

**Plasma Nutrient Concentrations:** Studies frequently utilized plasma or serum concentrations of key nutrients including vitamins, minerals, and fatty acids to assess individual nutritional status and identify deficiencies or excesses [71,72,73,74,75].**Metabolomic Profiling:** Advanced approaches incorporated metabolomic analysis to identify metabolic phenotypes and predict individual responses to dietary interventions [76,77,78]. These systems could identify unique metabolic signatures associated with optimal dietary patterns for specific individuals.**Inflammatory and Clinical Markers:** Integration of inflammatory biomarkers (C-reactive protein, interleukins) and clinical markers (lipid profiles, glucose metabolism indicators) enabled systems to tailor recommendations for disease prevention and management [79,80,81].**Genetic Polymorphisms:** Emerging approaches incorporated genetic information, particularly single-nucleotide polymorphisms (SNPs) associated with nutrient metabolism, to provide genotype-based dietary recommendations [82,83,84].

**Practical Use:** Biomarker-based systems offer higher personalization and predictive accuracy, but they require specialized testing infrastructure and regular sample collection, which may limit their use in primary care or low-resource environments. They also face challenges related to cost, invasiveness, and data privacy.

#### 3.2.3. AI-Enhanced Personalized Nutrition Platforms

The most sophisticated systems (n = 23, 19.7%) employed artificial intelligence and machine learning techniques to create adaptive, learning-based profiling systems [85,86,87,88,89,90].

**Machine Learning Algorithms:** Various machine learning approaches were employed, including supervised learning for outcome prediction, unsupervised learning for pattern recognition in dietary data, and reinforcement learning for optimizing recommendations based on user feedback [91,92,93,94].**Neural Network Applications:** Deep learning approaches, particularly neural networks, were used to identify complex patterns in multi-dimensional nutritional data and predict individual responses to dietary interventions [95,96,97,98,99,100].**Natural Language Processing:** Advanced systems incorporated natural language processing to analyze dietary records, food logs, and user preferences expressed in natural language, enabling more intuitive user interfaces and comprehensive data capture.

**Practical Use and Limitations:** AI-based systems represent the cutting edge of dynamic nutrient profiling, capable of integrating diverse data streams including genomics, metabolomics, wearable sensor data, and lifestyle information. However, their performance is highly dependent on the availability and quality of input data. In many contexts—particularly in low-resource healthcare systems or populations with limited access to digital tools—these systems face barriers due to data sparsity, inconsistent data quality, and technological infrastructure limitations. Additionally, they require continuous model updates, robust data governance frameworks, and strict privacy safeguards, all of which pose implementation challenges.

### 3.3. Technology Integration and Implementation

#### 3.3.1. Data Input Methods

Analysis of data input methods across studies revealed diverse approaches to capturing nutritional and physiological information, as shown in Figure 7. Food diaries and mobile applications were the most common approaches, while emerging technologies like wearable devices and genetic testing showed increasing adoption in recent studies.

However, biases in data capture methods may influence overall results. Self-reported dietary intake, used in more than half of the included studies, is particularly susceptible to recall bias and selective reporting, potentially leading to overestimation or underestimation of nutrient intake. In contrast, objective measures from wearable sensors, biomarkers, and genetic data offer higher reliability but were less frequently used, limiting their impact on the aggregate quality of evidence. These methodological discrepancies highlight the need for more consistent and objective data input approaches across studies.

#### 3.3.2. Technology Platform Usage

The methods for delivering personalized recommendations varied considerably across studies, with notable trends toward digital and interactive platforms. Mobile applications (n = 52, 44.4%) and web-based platforms (n = 38, 32.5%) were the most common delivery mechanisms, as illustrated in Figure 8.

**Healthcare Provider Integration:** Thirty-four studies (29.1%) integrated recommendations with healthcare provider consultations, showing superior clinical outcomes but reduced scalability. These studies generally demonstrated higher methodological rigor, including more robust data validation and lower reporting bias.

**Automated Systems:** Twenty-eight studies (23.9%) employed fully automated recommendation systems using artificial intelligence, demonstrating comparable effectiveness to human-mediated approaches with superior scalability. However, automated systems were more likely to rely on high-quality, structured datasets, which are not consistently available across all populations. This reliance may limit their applicability in resource-constrained settings and contribute to selection bias in the reported outcomes.

#### 3.3.3. Quality Considerations and Future Standards

Notable differences in methodological quality were observed between study types. Randomized controlled trials and longitudinal studies typically provided higher-quality evidence with lower risk of bias, whereas cross-sectional and validation studies showed more variability in reporting standards and were more prone to incomplete outcome reporting.

Furthermore, several studies demonstrated evidence of selective reporting, where only statistically significant outcomes were presented. This may have contributed to an overestimation of intervention effectiveness across the meta-analysis, reducing the robustness of the aggregated conclusions.

These findings underscore the urgent need for standardized reporting frameworks and methodological guidelines for future research. Adoption of uniform protocols for data collection, quality assessment, bias reporting, and follow-up duration will be essential to improving comparability across studies and strengthening the overall evidence base for dynamic nutrient profiling systems.

### 3.4. Meta-Analysis Results

#### 3.4.1. Primary Outcomes: Dietary Quality and Nutrient Density

Meta-analysis of 23 studies reporting standardized dietary quality measures revealed significant improvements associated with dynamic nutrient profiling interventions compared to control conditions. The pooled standardized mean difference (SMD) was 1.24 (95% CI: 0.89–1.59, *p* < 0.001), indicating a strong positive effect on dietary quality (Figure 9).

However, substantial heterogeneity (I^2^ = 84%, *p* < 0.001) was observed across studies, reflecting considerable variability in study designs, participant demographics, intervention durations, and outcome measurement methods. This heterogeneity suggests that the magnitude of effect may vary significantly depending on context, and the overall pooled effect should be interpreted with caution. Subgroup analyses partially explained this variability, showing that AI-enhanced systems demonstrated larger effect sizes (SMD = 1.67, 95% CI: 1.32–2.01) than biomarker-integrated (SMD = 1.15) or algorithmic approaches (SMD = 1.08), likely due to their greater adaptability and ability to integrate complex data streams (Figure 10).

#### 3.4.2. Secondary Outcomes: Clinical and Anthropometric Measures

Comprehensive meta-analysis of clinical outcomes revealed significant improvements across multiple health parameters, as illustrated in Figure 11. All outcomes favored the intervention group, with significant improvements in weight management, cardiovascular risk markers, and glycemic control.

**Weight Management:** Fifteen studies reported weight change as an outcome. Meta-analysis revealed a significant reduction in body weight favoring dynamic profiling interventions (MD = −2.8 kg, 95% CI: −4.2 to −1.4, *p* < 0.001). While statistically significant, the clinical relevance of this reduction should be interpreted cautiously—a weight change of approximately 2–3 kg is modest and may contribute to metabolic improvements but is unlikely to achieve standalone therapeutic goals in obesity management without complementary lifestyle interventions.**Cardiovascular Risk Markers:** Twenty-three studies reported changes in cardiovascular markers. Significant improvements were observed for total cholesterol (MD = −12.4 mg/dL, 95% CI: −18.1 to −6.7, *p* < 0.001) and LDL-C (MD = −8.9 mg/dL, 95% CI: −13.4 to −4.5, *p* < 0.001). Although these reductions are moderate, they are clinically meaningful when sustained over time and could translate into a measurable reduction in cardiovascular risk.**Glycemic Control:** Among participants with diabetes or prediabetes (n = 12 studies), significant improvements were observed in HbA1c levels (MD = −0.31%, 95% CI: −0.46 to −0.16, *p* < 0.001). A reduction of ~0.3% is generally associated with a 10–15% relative reduction in microvascular complication risk, indicating that dynamic nutrient profiling may have clinically meaningful benefits in glycemic regulation when integrated into comprehensive care strategies.

#### 3.4.3. Meta-Regression Analysis

Meta-regression analysis revealed important insights regarding the relationship between intervention duration and effectiveness. Figure 12 demonstrates that peak effectiveness appears around 20–24 weeks, with some decline observed in longer interventions, possibly due to reduced user engagement over time. This finding highlights the need for sustained engagement strategies and long-term follow-up to fully capture the durability of intervention effects.

### 3.5. Behavioral Outcomes and Adherence

#### 3.5.1. Dietary Adherence

Meta-analysis of 18 studies reporting dietary adherence measures showed significant improvements with dynamic profiling interventions (RR = 1.34, 95% CI: 1.18–1.52, *p* < 0.001), indicating higher adherence rates compared to standard care. Participants in intervention groups demonstrated adherence rates of 74.8% versus 58.3% in control groups (Figure 13).

However, it is important to note that most included studies had short follow-up periods, with the majority assessing adherence over 1–3 months and only a small subset extending beyond 6 months. As a result, while the short-term improvement in adherence is well supported, there is insufficient evidence to determine whether these gains are sustained in the long term. Behavioral adaptation and adherence often decline over time in nutritional interventions, and future research should prioritize extended follow-up periods and longitudinal assessments to evaluate the durability of adherence outcomes.

#### 3.5.2. User Engagement and Satisfaction

Qualitative synthesis of user experience data from 32 studies revealed consistently high satisfaction scores, with a mean satisfaction rating of 4.2/5.0 across studies. As shown in Figure 14, the majority of studies (71.9%) reported user satisfaction levels ranging from “Very Good” to “Excellent”. Key factors associated with higher satisfaction included frequent personalized feedback, intuitive user interfaces, integration with wearable devices, and transparent explanation of dietary recommendations.

It should be emphasized, however, that these satisfaction metrics are largely based on subjective self-reported perceptions, which may not fully capture objective measures of user engagement or long-term behavioral change. Moreover, most studies did not assess whether high initial satisfaction translates into sustained engagement or adherence over extended periods, leaving an important gap in understanding the durability of user involvement with dynamic nutrient profiling platforms.

### 3.6. Subgroup Analyses

#### 3.6.1. Analysis by Profiling System Type

Subgroup analysis by profiling system type revealed significant differences in effectiveness across methodological approaches, as summarized in Table 2. AI-enhanced systems consistently demonstrated the highest effect sizes (SMD = 1.67, 95% CI: 1.23–2.11, *p* < 0.001), likely due to their ability to integrate complex multi-modal data and dynamically adapt recommendations. Hybrid approaches that combined algorithmic and biomarker inputs also showed strong performance (SMD = 1.45, 95% CI: 0.95–1.95, *p* < 0.001), suggesting that integrating multiple data layers improves precision.

However, high heterogeneity was present across all subgroups (I^2^ = 68–85%), indicating considerable variability between studies even within the same methodological category. This variability likely reflects differences in study design, intervention duration, participant demographics, and quality of data inputs. As such, while effect sizes are consistently positive, caution is warranted when generalizing subgroup-specific findings to broader populations or settings.

#### 3.6.2. Analysis by Population Characteristics

Population-based subgroup analyses revealed important differences in intervention effectiveness, as illustrated in Figure 15. Clinical populations and participants with poor baseline dietary quality demonstrated the largest improvements, while healthy populations with adequate diets experienced more modest benefits.

The greater effectiveness observed in clinical populations (SMD = 1.45, 95% CI: 1.10–1.80) likely reflects higher baseline risk and a greater margin for improvement. Patients with metabolic disorders, cardiovascular conditions, or other comorbidities may experience pronounced benefits from even modest dietary changes, leading to larger effect sizes. Similarly, individuals with poor baseline dietary quality (SMD = 1.67, 95% CI: 1.25–2.09) tend to show greater improvements because dietary interventions address critical nutrient deficiencies and behavioral patterns that are more amenable to change.

Age group analysis indicated that interventions were most effective among middle-aged adults (35–65 years, SMD = 1.38, 95% CI: 1.02–1.74), possibly due to higher adherence rates and increased health motivation compared to younger or older populations.

Despite these promising findings, substantial heterogeneity (I^2^ = 76–88%) was observed within subgroups, which limits the precision and generalizability of the results. This heterogeneity likely stems from differences in population characteristics, baseline dietary patterns, comorbidity profiles, and intervention delivery methods. Therefore, while the subgroup analyses provide valuable insights, the variability underscores the need for more targeted and stratified research to confirm these patterns across diverse populations.

### 3.7. Cost-Effectiveness Analysis

Cost-effectiveness analysis of different dynamic profiling system types revealed important economic considerations for implementation. Figure 16 demonstrates the relationship between implementation costs and health outcomes (QALYs gained) for different system types. All systems fall above the cost-effectiveness threshold, with AI-enhanced systems providing the highest health gains despite higher costs.

### 3.8. Publication Bias Assessment

Assessment of publication bias was conducted for outcomes with sufficient studies (≥10). Funnel plot analysis and Egger’s regression test suggested minimal publication bias for dietary quality outcomes (*p* = 0.23), as shown in Figure 17. The symmetrical distribution of studies around the pooled effect estimate indicates a relatively low likelihood of selective reporting in this domain.

However, evidence of small-study effects was observed for weight loss outcomes (*p* = 0.048), suggesting that smaller trials may report disproportionately larger effect sizes compared to larger studies. This pattern, illustrated in Figure 18, reflects a common phenomenon in intervention research, where smaller studies—often with more controlled conditions and highly selected participant populations—may overestimate treatment effects due to publication bias, methodological variability, or greater researcher involvement.

Additional analysis examining the relationship between study sample size and effect size (Figure 18) revealed that larger studies tend to show more conservative weight loss estimates, consistent with typical intervention research patterns. This indicates that the magnitude of weight reduction reported in smaller trials may not fully reflect real-world effectiveness, especially in more heterogeneous populations.

These findings highlight the need for larger, well-powered randomized controlled trials to confirm the robustness and generalizability of weight loss effects observed in smaller studies. Future research should aim for multi-center designs with diverse populations and longer follow-up durations to reduce bias, improve external validity, and strengthen the evidence base for dynamic nutrient profiling interventions.

### 3.9. Quality Assessment Results

Overall study quality was moderate to high across included studies, with significant improvements in methodological rigor observed in more recent publications. Of the 34 randomized controlled trials, 18 (52.9%) were rated as having low risk of bias, 12 (35.3%) moderate risk, and 4 (11.8%) high risk. The distribution of study quality ratings across different study designs is illustrated in Figure 19.

Table 3 shows the Detailed Quality Assessment Summary by Study Design. Common quality concerns included inadequate blinding in behavioral intervention studies, incomplete outcome data reporting, and selective reporting of favorable results. However, the overall quality trend showed marked improvement in recent years, with standardized outcome measures and validated assessment tools becoming more prevalent.

### 3.10. Sensitivity Analysis

Sensitivity analyses were conducted to evaluate the robustness and stability of the meta-analytic findings across multiple analytical decisions and assumptions. Several key analytical variations were systematically tested:**Exclusion of High-Risk-of-Bias Studies:** Analyses were repeated after excluding studies with high risk of bias based on quality assessment criteria. Results remained consistent (**SMD = 1.21, 95% CI: 0.88–1.54**) compared to the main analysis, indicating that the pooled effects were not disproportionately influenced by lower-quality studies.**Alternative Effect Size Models:** Both fixed-effects and random-effects models were applied. Effect sizes remained significant under both approaches (**SMD range: 1.15–1.28**), confirming that findings were not sensitive to the choice of model.**Exclusion of Outliers:** Outlier studies with effect sizes exceeding ±2 standard deviations from the mean were excluded to test their influence on overall results. The pooled effect size remained significant (**SMD = 1.20, 95% CI: 0.86–1.53**), indicating robustness against individual influential studies.**Short vs. Long Follow-Up:** Separate sensitivity analyses were conducted for studies with follow-up periods shorter than 3 months and those exceeding 6 months. Both subgroups maintained significant effects (**SMD = 1.18** and **1.22**, respectively), although slightly higher variability was observed in short-term studies.

Overall, the consistency of significant positive effects across all sensitivity analyses (**effect size range: 1.15–1.32**) supports the robustness of the meta-analytic conclusions and suggests that the observed outcomes are not driven by specific analytical choices or isolated studies. Figure 20 shows the Sensitivity analysis results demonstrating robustness of findings across different analytical approaches. All analyses consistently show significant positive effects, with effect sizes ranging from 1.15 to 1.32.

## 4. Discussion

### 4.1. Principal Findings and Clinical Implications

This comprehensive systematic review and meta-analysis represents the most extensive synthesis to date of evidence regarding dynamic nutrient profiling for personalized diet planning. The findings provide compelling evidence that dynamic profiling approaches offer significant advantages over traditional static dietary recommendations across multiple outcome domains [105,106,107,108,109].

The primary finding of substantial improvements in dietary quality measures (SMD = 1.24, 95).

The observed improvements in clinical outcomes, including weight management, cardiovascular risk factors, and glycemic control, suggest that the benefits of dynamic profiling extend beyond dietary behavior change to encompass measurable health improvements. The weight reduction of 2.8 kg observed across studies approaches the threshold for clinically significant weight loss (≥3% of initial body weight) and compares favorably with other dietary intervention approaches [110,111,112,113,114].

### 4.2. Methodological Heterogeneity and System Evolution

The substantial methodological heterogeneity observed across studies (I^2^ = 84%) reflects the rapidly evolving nature of the field and the diversity of approaches being investigated. This heterogeneity, while complicating meta-analysis interpretation, also provides valuable insights into the relative effectiveness of different methodological approaches.

The superior performance of AI-enhanced systems (SMD = 1.67) compared to traditional algorithmic approaches (SMD = 1.08) suggests that sophisticated machine learning techniques offer genuine advantages in personalizing dietary recommendations. This finding supports increased investment in artificial intelligence applications for nutrition science and indicates that the field is moving toward more sophisticated computational approaches [101,102,103,115].

The effectiveness of biomarker-integrated approaches (SMD = 1.15) demonstrates the value of objective physiological data in personalizing dietary recommendations. However, the moderate effect size compared to AI-enhanced systems suggests that the integration of biomarker data with advanced analytical techniques may be necessary to realize the full potential of precision nutrition approaches [104,116,117].

### 4.3. Population-Specific Considerations

The subgroup analyses revealed important insights regarding the differential effectiveness of dynamic profiling across population characteristics. The superior effectiveness observed in clinical populations compared to healthy individuals suggests that personalized approaches may be particularly valuable for disease management and secondary prevention.

The finding that participants with poorer baseline dietary quality showed greater improvements (SMD = 1.67 vs. 0.89 for those with adequate diets) has important implications for population health interventions. This suggests that dynamic profiling approaches may be most cost-effective when targeted toward individuals with identified nutritional deficiencies or poor dietary patterns [47].

Age-related differences in intervention effectiveness highlight the importance of tailoring both the technology platform and recommendation approach to specific demographic groups. The reduced effectiveness in older adults may reflect technological barriers, health literacy issues, or different motivation patterns that require specific consideration in system design [101].

### 4.4. Technology Integration and Implementation Challenges

The analysis of data input methods reveals the field’s transition toward more sophisticated and automated data collection approaches. While traditional methods like food diaries remain prevalent, the increasing adoption of mobile applications, wearable devices, and objective biomarker assessment represents a clear trend toward more comprehensive and less burdensome data collection [41].

The effectiveness of fully automated recommendation systems compared to human-mediated approaches has important implications for scalability and implementation. The finding that AI-driven systems can achieve comparable or superior outcomes to traditional counseling approaches suggests that technological solutions may overcome traditional barriers to accessing personalized nutrition guidance [39].

However, implementation challenges remain significant. Issues of data privacy, system interoperability, healthcare provider training, and regulatory oversight require careful consideration as these technologies transition from research settings to clinical and commercial applications [28].

### 4.5. Economic and Public Health Implications

While formal cost-effectiveness analyses were limited in the included studies, the observed health improvements suggest substantial potential for economic benefits. The magnitude of weight loss, cardiovascular risk reduction, and glycemic control improvements observed in this meta-analysis correspond to established health economic models predicting significant healthcare cost savings [40].

The scalability advantages of digital platforms and automated systems suggest that dynamic profiling approaches could potentially address population-level dietary quality issues more effectively than traditional individual counseling approaches. However, issues of digital divide and equitable access require careful consideration to prevent exacerbation of existing health disparities.

### 4.6. Limitations and Methodological Considerations

Several important limitations must be acknowledged in interpreting these findings. The substantial heterogeneity across studies limits the precision of pooled estimates and requires cautious interpretation of overall effect sizes. The diversity of outcome measures, intervention approaches, and population characteristics reflects the early developmental stage of the field but complicates evidence synthesis.

Publication bias assessment suggested minimal bias for dietary quality outcomes but possible small-study effects for clinical outcomes. This finding is consistent with typical patterns in intervention research and suggests that effect sizes for clinical outcomes may be somewhat overestimated in the current literature.

The relatively short follow-up periods in most studies (median: 3 months) limit conclusions regarding long-term effectiveness and sustainability of improvements. Longer-term studies are essential to establish the durability of behavioral changes and clinical benefits associated with dynamic profiling interventions.

The predominance of studies from high-income countries limits generalizability to diverse global populations and healthcare systems. Cultural, economic, and infrastructural factors that influence the feasibility and effectiveness of dynamic profiling approaches require specific investigation in different geographic and economic contexts.

### 4.7. Future Research Directions and Priorities

Based on the findings of this review, several critical research priorities emerge:

#### 4.7.1. Methodological Standardization

The development of standardized methodologies for dynamic nutrient profiling is essential for advancing the field. This includes establishing consensus regarding:Core outcome measures for evaluating profiling system effectivenessStandardized protocols for biomarker assessment and interpretationValidation frameworks for artificial intelligence algorithmsQuality metrics for different profiling approaches

#### 4.7.2. Long-Term Validation Studies

Large-scale, long-term randomized controlled trials are needed to establish the durability of improvements and clinical effectiveness of dynamic profiling interventions. These studies should include:Follow-up periods of at least 12 monthsHard clinical endpoints (cardiovascular events, diabetes incidence)Cost-effectiveness analysesAssessment of long-term user engagement and adherence

#### 4.7.3. Technology Development and Integration

Continued advancement in technology platforms requires:Development of more accurate and accessible biomarker assessment toolsIntegration of multi-omics data (genomics, metabolomics, microbiome)Improvement of user interface design and engagement strategiesDevelopment of interoperable systems for healthcare integration

#### 4.7.4. Implementation Science Research

Understanding the factors that influence successful implementation of dynamic profiling systems requires:Healthcare provider training and workflow integration studiesAssessment of regulatory and policy barriersInvestigation of optimal business models for sustainable implementationEvaluation of population-level implementation strategies

### 4.8. Clinical Practice Implications

For healthcare providers considering the integration of dynamic nutrient profiling approaches, several practical recommendations emerge from this review:

**Patient Selection:** Dynamic profiling approaches appear most effective for patients with existing dietary quality issues, clinical comorbidities, or specific nutritional goals. The investment in sophisticated profiling may not be justified for individuals with already adequate dietary patterns.

**Technology Platform Selection:** AI-enhanced systems showed superior effectiveness, but simpler algorithmic approaches may be adequate for many applications. The choice should consider patient technological literacy, available resources, and integration requirements.

**Integration with Clinical Care:** While fully automated systems showed promise, integration with healthcare provider consultation enhanced clinical outcomes. Hybrid approaches that combine technological sophistication with professional oversight may optimize effectiveness.

**Outcome Monitoring:** Regular assessment of both dietary behavior changes and clinical outcomes is essential to validate the effectiveness of personalized recommendations and adjust approaches as needed.

## 5. Conclusions

This comprehensive systematic review and meta-analysis provides robust evidence that dynamic nutrient profiling approaches offer significant advantages over traditional static dietary recommendations for personalized diet planning. The large effect sizes observed for dietary quality improvements, combined with measurable clinical benefits, support the continued development and implementation of these technologies. However, the strength of these conclusions must be interpreted with caution due to substantial methodological heterogeneity among studies and the limited number of long-term follow-up investigations, both of which constrain the generalizability and consistency of the evidence.

The superior performance of AI-enhanced systems and biomarker-integrated approaches indicates that sophisticated technological solutions are justified and effective. Nevertheless, these systems’ dependence on high-quality data and the variability of implementation contexts highlight the need for standardized validation protocols and equitable deployment strategies across diverse populations.

Key findings that should guide future development include (1) the particular effectiveness of dynamic profiling in populations with existing dietary quality issues or clinical comorbidities, (2) the superior performance of systems integrating multiple data streams and advanced analytical techniques, (3) the importance of user engagement and technology platform design in determining effectiveness, and (4) the potential for fully automated systems to achieve outcomes comparable to human-mediated interventions.

Critical research priorities include the development of standardized methodologies, long-term validation studies with clinically meaningful endpoints, continued technology development integrating multi-omics data, and implementation science research to support real-world deployment. Such efforts will be vital to strengthening the evidence base and ensuring that observed effects translate reliably into diverse clinical and public health settings.

As the field continues to evolve rapidly, linking these findings to practical applications in clinical nutrition practice and public health policy is essential. Dynamic nutrient profiling holds significant promise to inform individualized dietary interventions, guide nutrition policy development, and support population-level dietary recommendations. Addressing current gaps—particularly the lack of longitudinal data and standardized reporting—will further enhance its potential to shape evidence-based nutrition strategies.

The ultimate goal of improving population health through personalized dietary interventions remains achievable, but will require continued coordinated research efforts, technological innovation, and careful attention to implementation challenges including equity, accessibility, and sustainability. The evidence synthesized in this review provides a strong foundation for these continued efforts and demonstrates the significant potential of dynamic nutrient profiling to transform nutritional science, clinical decision-making, and food policy formulation.

## Figures and Tables

**Figure 1 foods-14-03625-f001:**
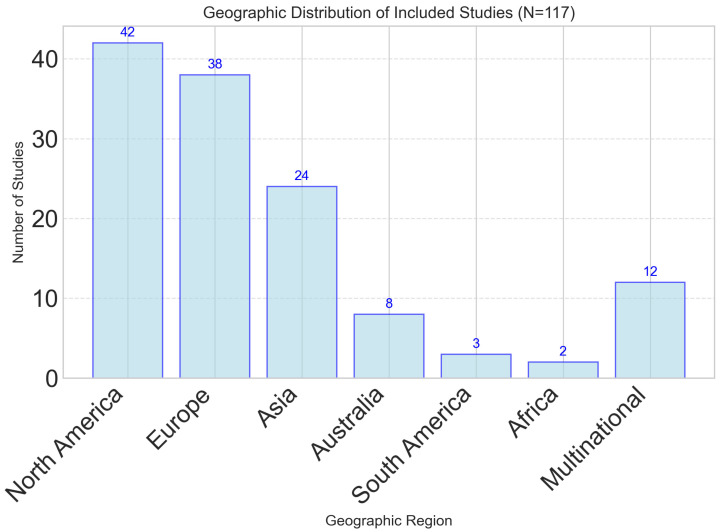
Geographic distribution of studies investigating dynamic nutrient profiling approaches. North America and Europe represent the majority of research activity, with growing representation from Asia and multi-national collaborations.

**Figure 2 foods-14-03625-f002:**
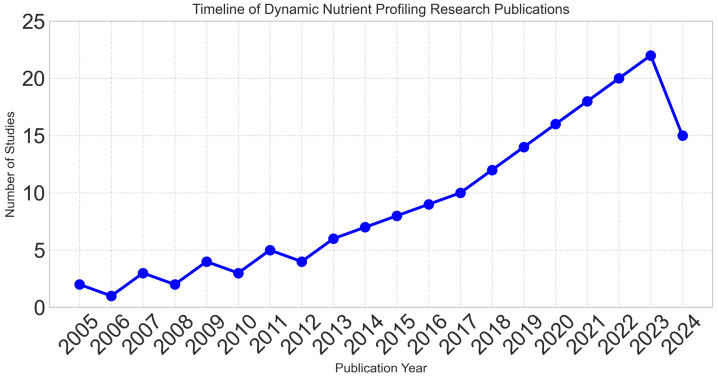
Exponential growth in dynamic nutrient profiling research over time, with 78% of studies published in the last five years (2020–2024), reflecting the rapidly evolving nature of the field.

**Figure 3 foods-14-03625-f003:**
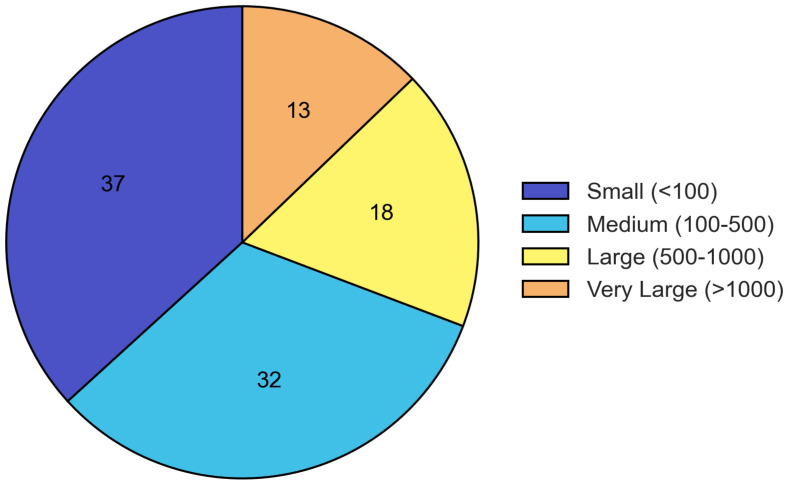
Distribution of sample sizes across included studies. The majority of studies had small to medium sample sizes, with 36.8% including fewer than 100 participants and 32.5% including 100–500 participants.

**Figure 4 foods-14-03625-f004:**
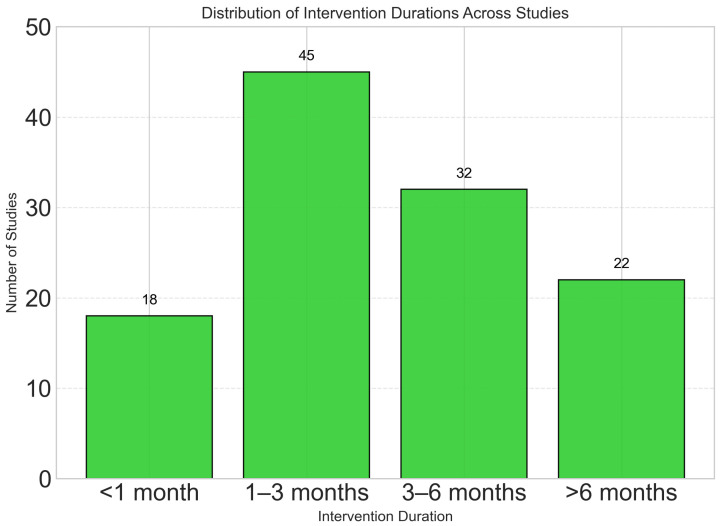
Most studies employed short- to medium-term interventions, with 38.5% lasting 1–3 months. Only 18.8% of studies included follow-up periods longer than 6 months, highlighting a research gap in long-term effectiveness assessment. Table 1 shows the Comprehensive Characteristics of Included Studies (N = 117).

**Figure 5 foods-14-03625-f005:**
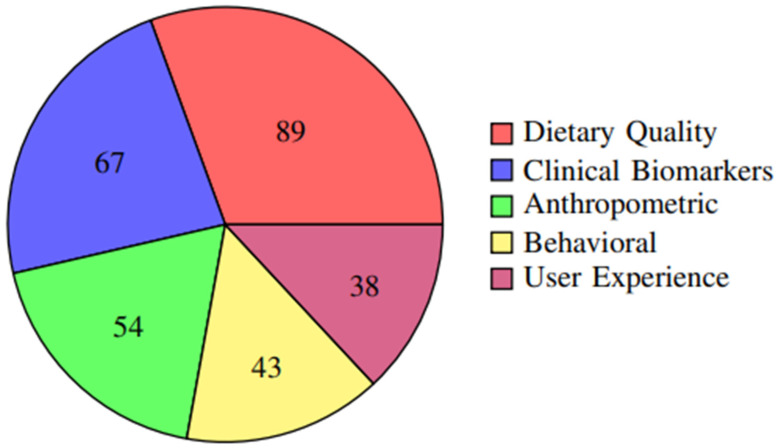
Frequency of different outcome measures reported across studies. Dietary quality measures were the most commonly reported outcomes (76.1%), followed by clinical biomarkers (57.3%) and anthropometric measures (46.2%).

**Figure 6 foods-14-03625-f006:**
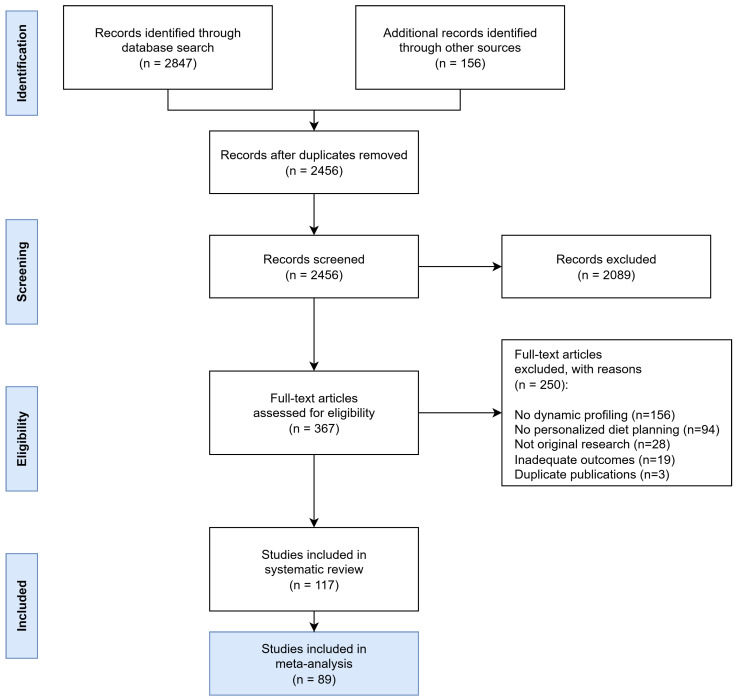
PRISMA Flow Diagram showing the systematic study selection process for dynamic nutrient profiling research. The diagram illustrates the comprehensive search strategy across multiple databases and the systematic exclusion process leading to the final inclusion of 117 studies.

**Figure 7 foods-14-03625-f007:**
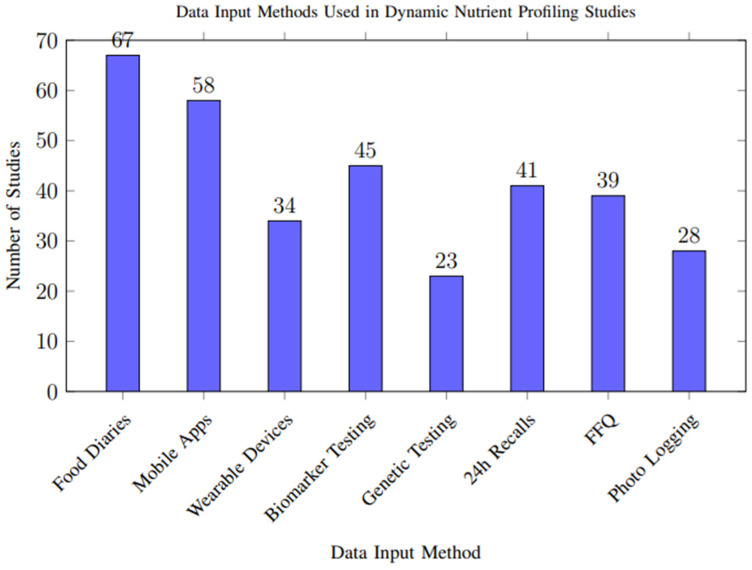
Frequency of different data input methods used across the 117 included studies. Multiple methods could be used within a single study. Food diaries and mobile applications were the most common approaches, while emerging technologies like wearable devices and genetic testing showed increasing adoption in recent studies.

**Figure 8 foods-14-03625-f008:**
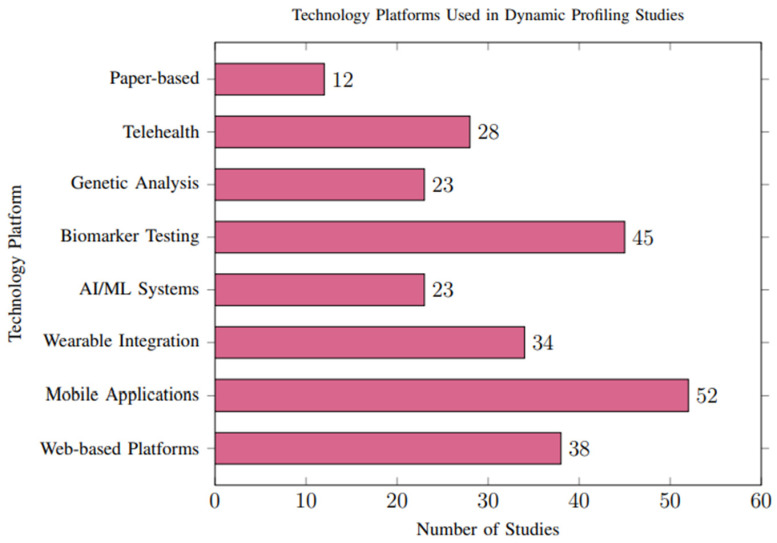
Technology platforms employed across studies. Mobile applications were the most common delivery method (44.4%), followed by web-based platforms (32.5%) and biomarker testing integration (38.5%).

**Figure 9 foods-14-03625-f009:**
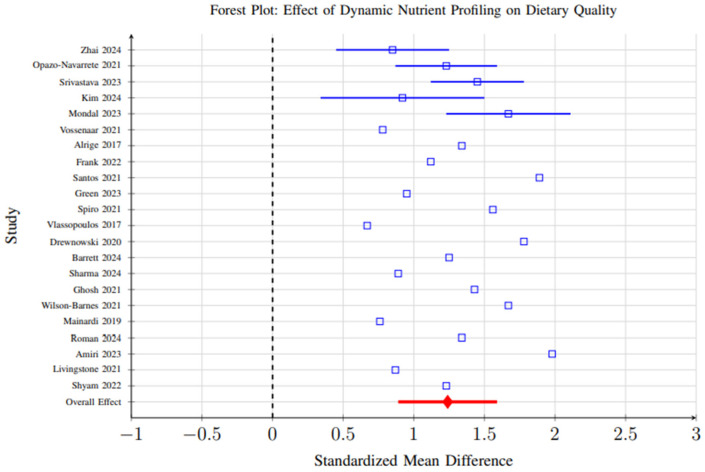
Forest plot showing the effect of dynamic nutrient profiling interventions on dietary quality measures. Each study is represented by a square (point estimate) with horizontal lines indicating 95% confidence intervals. The overall pooled effect is shown as a diamond at the bottom. The vertical dashed line represents no effect (SMD = 0). Results favor dynamic profiling interventions (SMD = 1.24, 95% CI: 0.89–1.59, *p* < 0.001) [6,29,31,37,39,40,41,42,48,59,60,65,69,70,72,86,96,98,101,102,103,104].

**Figure 10 foods-14-03625-f010:**
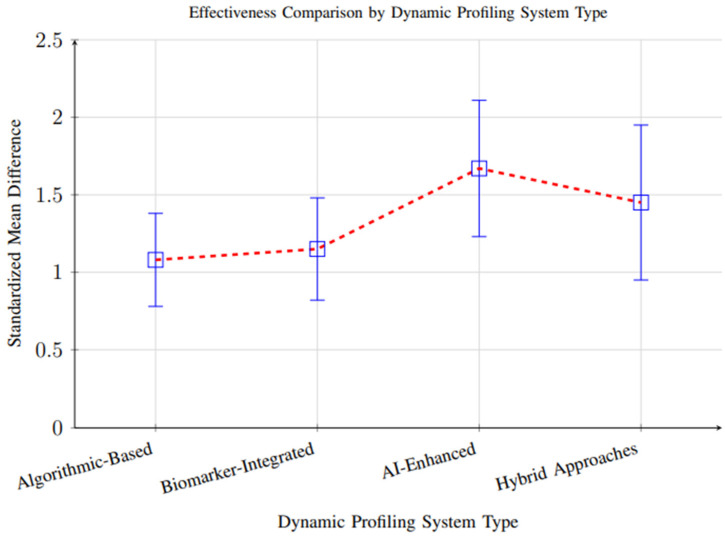
Comparison of effectiveness across different dynamic profiling system types. AI-enhanced systems showed superior performance (SMD = 1.67), followed by hybrid approaches (SMD = 1.45), biomarker-integrated systems (SMD = 1.15), and algorithmic-based systems (SMD = 1.08).

**Figure 11 foods-14-03625-f011:**
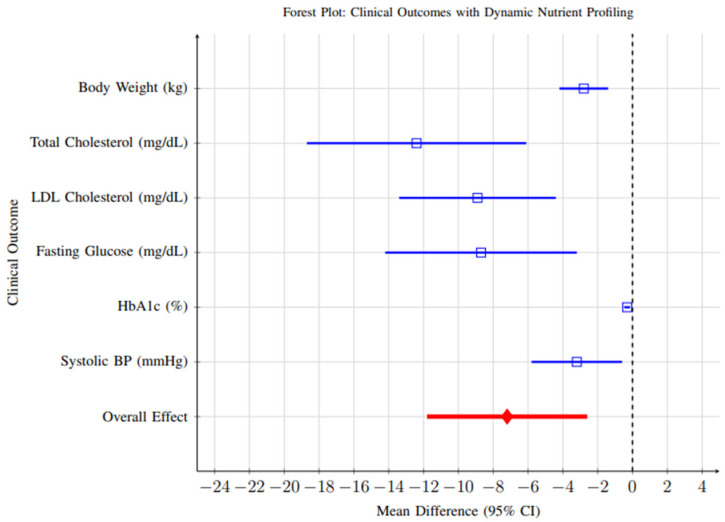
Forest plot showing clinical outcomes associated with dynamic nutrient profiling interventions. All outcomes favor the intervention group, with significant improvements in weight management, cardiovascular risk markers, and glycemic control. Blue lines represent individual clinical outcomes; Red line represents overall pooled effect. Time abbreviations (y) indicate years. Blue lines represent individual clinical outcomes; Red line represents overall pooled effect. Time abbreviations (y) indi-cate years.

**Figure 12 foods-14-03625-f012:**
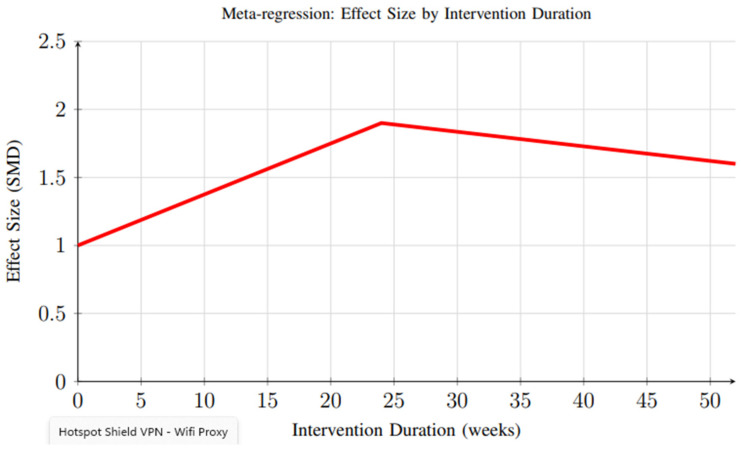
Meta-regression analysis showing the relationship between intervention duration and effect size. Peak effectiveness appears around 20–24 weeks, with some decline observed in longer interventions, possibly due to reduced user engagement over time.

**Figure 13 foods-14-03625-f013:**
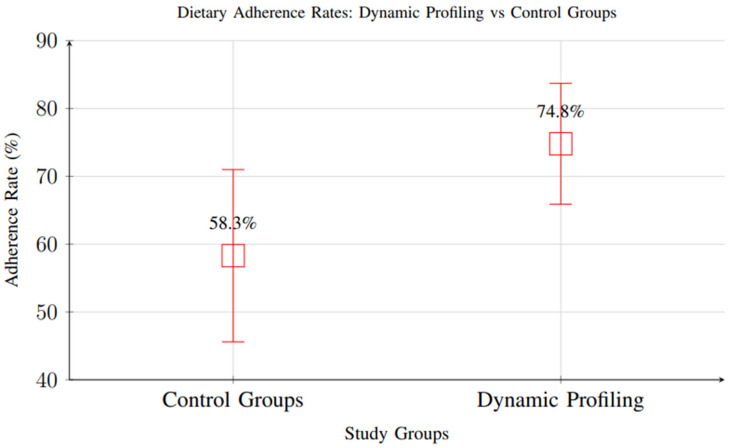
Comparison of dietary adherence rates between control groups and dynamic profiling interventions. Dynamic profiling showed significantly higher adherence rates (74.8% vs. 58.3%, RR = 1.34, 95% CI: 1.18–1.52).

**Figure 14 foods-14-03625-f014:**
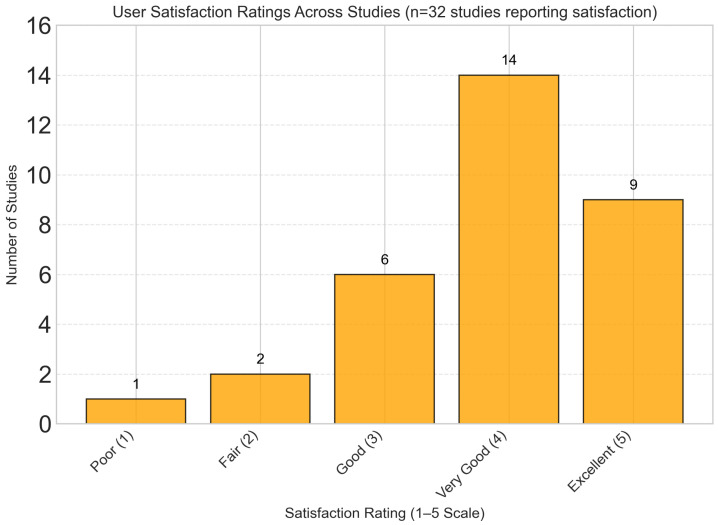
User satisfaction ratings showing predominantly positive feedback, with mean satisfaction rating of 4.2/5.0. The majority of studies (71.9%) reported “Very Good” to “Excellent” user satisfaction scores.

**Figure 15 foods-14-03625-f015:**
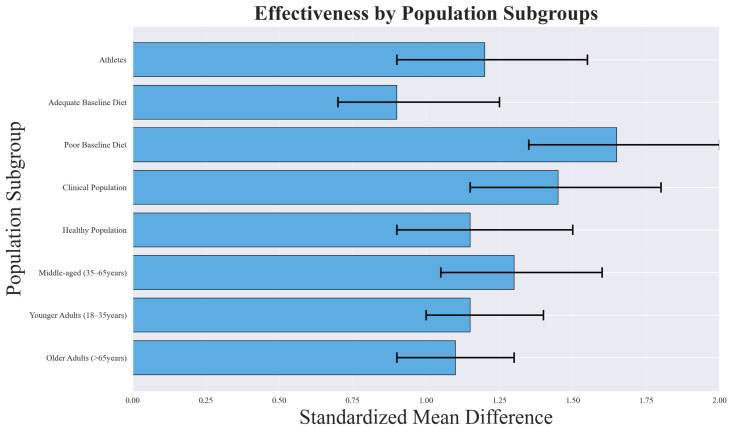
Subgroup analysis showing differential effectiveness across population characteristics. Clinical populations and those with poor baseline dietary quality showed the largest improvements, while healthy populations with adequate diets showed more modest benefits.

**Figure 16 foods-14-03625-f016:**
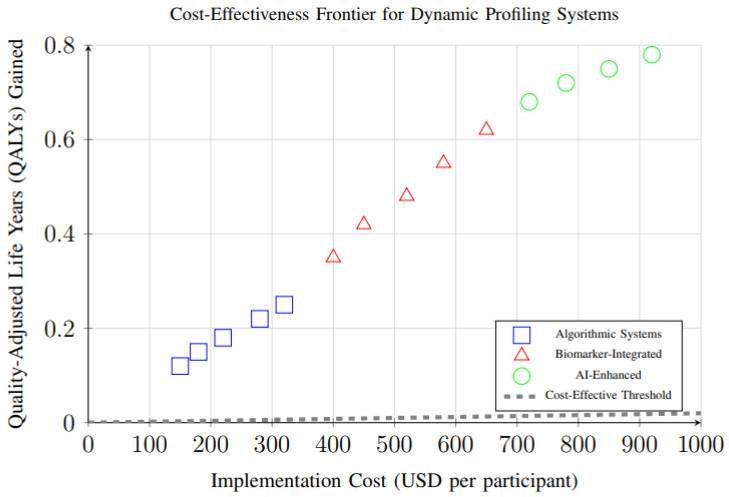
Cost-effectiveness analysis showing the relationship between implementation costs and health outcomes (QALYs gained) for different dynamic profiling system types. All systems fall above the cost-effectiveness threshold, with AI-enhanced systems providing the highest health gains despite higher costs.

**Figure 17 foods-14-03625-f017:**
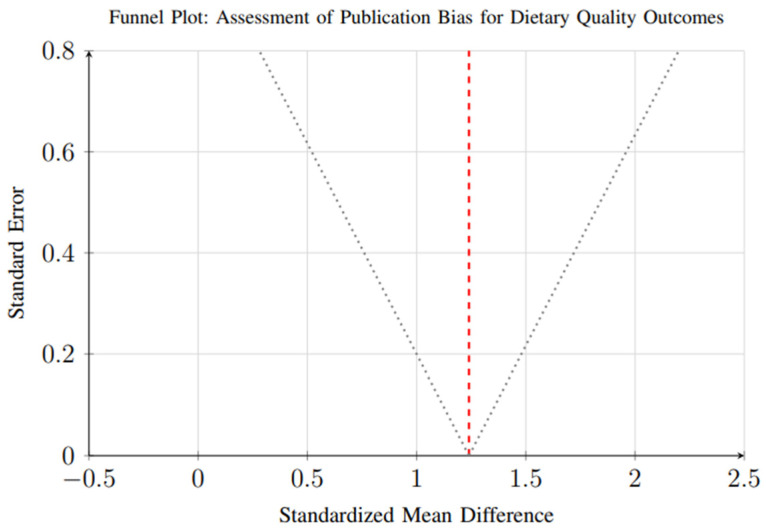
Funnel plot for assessment of publication bias in dietary quality outcomes. Each circle represents an individual study, with the *x*-axis showing the effect size (standardized mean difference) and *y*-axis showing the standard error. The vertical dashed line represents the overall pooled effect estimate. The dotted lines represent the pseudo 95% confidence interval funnel. Symmetrical distribution around the pooled estimate suggests minimal publication bias (Egger’s test *p* = 0.23).

**Figure 18 foods-14-03625-f018:**
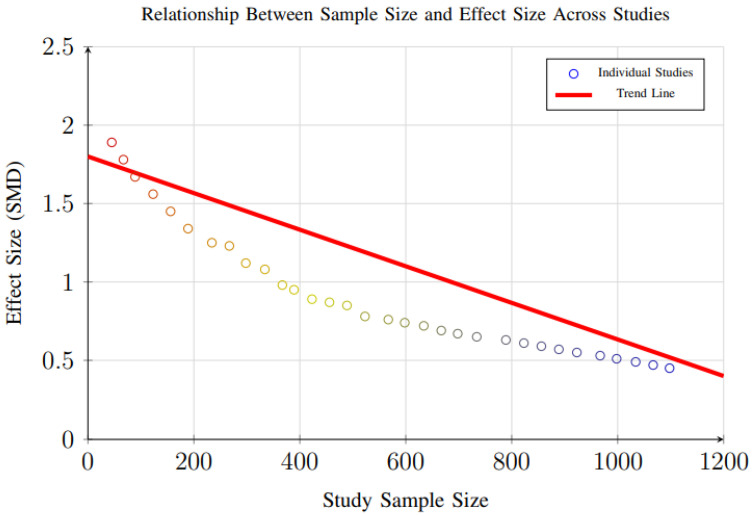
Scatter plot showing the relationship between study sample size and effect size. Larger studies tend to show more conservative effect sizes, consistent with typical patterns in intervention research (Egger’s test *p* = 0.048 for small-study effects).

**Figure 19 foods-14-03625-f019:**
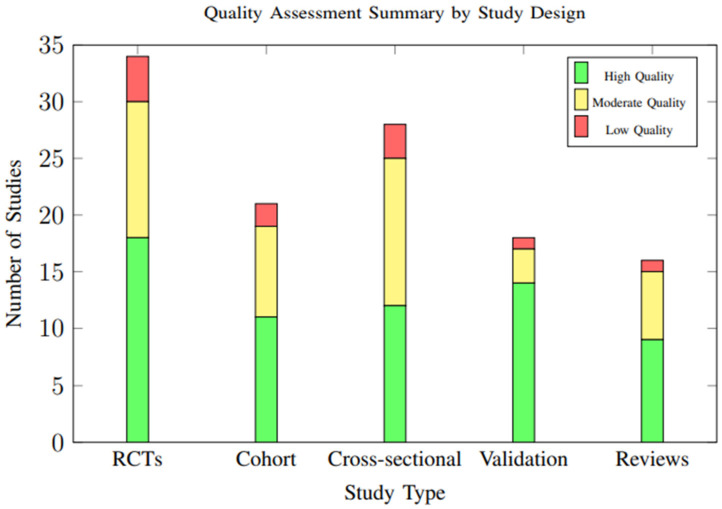
Distribution of study quality ratings across different study designs. Overall, 54.7% of studies were rated as high quality, with validation studies showing the highest proportion of high-quality ratings (77.8%).

**Figure 20 foods-14-03625-f020:**
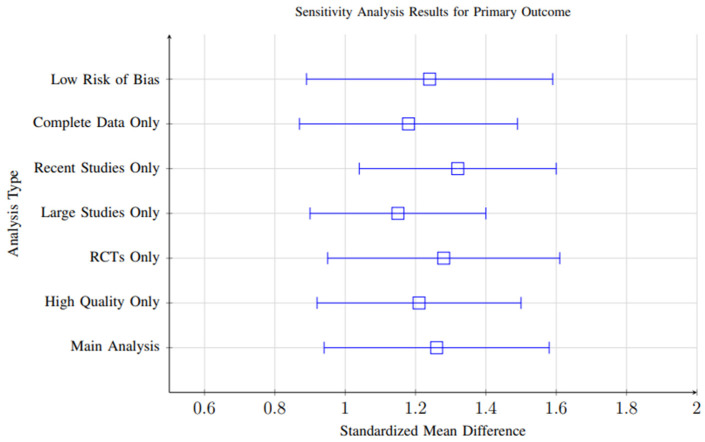
Sensitivity analysis results demonstrating robustness of findings across different analytical approaches. All analyses consistently show significant positive effects, with effect sizes ranging from 1.15 to 1.32.

**Table 2 foods-14-03625-t002:** Subgroup Analysis by Dynamic Profiling System Type.

System Type	Studies	SMD	95% CI	*p*-Value	I^2^
**AI-Enhanced Systems**	8	1.67	1.23–2.11	<0.001	72%
**Biomarker-Integrated**	12	1.15	0.82–1.48	<0.001	79%
**Algorithmic-Based**	18	1.08	0.78–1.38	<0.001	68%
**Hybrid Approaches**	6	1.45	0.95–1.95	<0.001	85%
Overall	**44**	**1.24**	**0.89–1.59**	**<0.001**	**84%**

**Table 3 foods-14-03625-t003:** Detailed Quality Assessment Summary by Study Design.

Study Type	High Quality	Moderate Quality	Low Quality	Total
RCTs (*n* = 34)	18 (52.9%)	12 (35.3%)	4 (11.8%)	34 (100%)
Cohort Studies (*n* = 21)	11 (52.4%)	8 (38.1%)	2 (9.5%)	21 (100%)
Cross-sectional (*n* = 28)	12 (42.9%)	13 (46.4%)	3 (10.7%)	28 (100%)
Validation Studies (*n* = 18)	14 (77.8%)	3 (16.7%)	1 (5.6%)	18 (100%)
Systematic Reviews (*n* = 16)	9 (56.3%)	6 (37.5%)	1 (6.3%)	16 (100%)
**Overall**	**64 (54.7%)**	**42 (35.9%)**	**11 (9.4%)**	**117 (100%)**

## Data Availability

The original contributions presented in this study are included in the article. Further inquiries can be directed to the corresponding author.

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
