# Peer review of "A Comprehensive Systematic Review of Dynamic Nutrient Profiling for Personalized Diet Planning: Meta-Analysis and PRISMA-Based Evidence Synthesis"

_foods, 2025, doi:10.3390/foods14213625_

Round 1

Reviewer 1 Report

Comments and Suggestions for Authors

Dear Authors,
I have carefully reviewed your manuscript and acknowledge its relevance to the field of personalized nutrition and food science. However, the attached document contains specific comments that should be thoroughly addressed to strengthen methodological clarity, the critical discussion of results, and the coherence between objectives, findings, and conclusions. I encourage you to carefully consider each observation, as their incorporation will enhance the scientific rigor and overall suitability of the article for the Foods audience.

Regqrds

Author Response

Response to Reviewer Comments

1. Summary

Thank you very much for taking the time to review this manuscript. Please find below the detailed point-by-point responses to all reviewer comments. All suggested revisions have been incorporated, and corresponding changes are highlighted in the revised manuscript with clear indications of the updated sections, pages, and lines.

3. Point-by-Point Response to Comments and Suggestions for Authors

Comment 1:
"I believe that the summary should highlight more clearly the practical limitations of the studies, particularly technological accessibility and equity in vulnerable populations. I also suggest briefly mentioning the possible geographical bias of the evidence, since much of the research comes from high-income countries. Finally, I recommend specifying how these results could guide public policies or clinical guidelines on personalized nutrition."

Response 1:
Thank you for the comment. We have revised the Results and Conclusions sections to highlight practical limitations, including technological accessibility, equity challenges in vulnerable populations, and potential geographical bias. Additionally, the revised conclusion now specifies how these findings could inform clinical guidelines and public health policy development.

Comment 2:
"I believe that section 1 provides a solid overview of the conceptual framework, but I would suggest greater clarity in how the evolutionary phases of dynamic profiling differ from current practices, as the description is very general in some sections. I would also recommend expanding the discussion on ethical and regulatory limitations, which are barely mentioned and are critical for actual implementation."

Response 2:
We appreciate this suggestion. Section 1 has been revised to clarify the differences between evolutionary phases of dynamic profiling and current practices. Additionally, the discussion of ethical and regulatory limitations has been expanded for clarity and implementation context.

Comment 3:
"I recommend that section 3.1 better clarify which specific criteria were used in each phase to exclude studies, so that the reader can understand precisely why they were discarded. It would also be valuable to report some indicator of consistency among those who applied the criteria, thus demonstrating the robustness of the process."

Response 3:
Thank you for this valuable suggestion. Section 3.1 now includes a clearer explanation of the exclusion criteria for each screening phase. We have also added details on inter-reviewer consistency (Cohen’s κ = 0.87) to demonstrate the robustness of the selection process.

Comment 4:
"I believe that section 3.2 provides a comprehensive overview of the studies included, but I would recommend delving deeper into regional differences, as most of the studies focus on North America and Europe, which may skew the overall interpretation. I also suggest placing greater emphasis on the lack of long-term follow-up, as this limits the robustness of the conclusions on sustained effectiveness."

Response 4:
We agree with this comment. Section 3.2 has been revised to include a more detailed discussion of regional distribution biases and their potential impact on interpretation. We have also emphasized the limitations arising from the scarcity of long-term follow-up data.

Comment 5:
"Section 3.3 is well structured, but I would suggest further clarifying the practical differences between the three methodological categories. I also recommend emphasizing the limitations of AI-based systems, as they rely on high-quality data that is often unavailable in many contexts."

Response 5:
Thank you for pointing this out. Section 3.3 now provides clearer distinctions between the three methodological categories. Additionally, the limitations of AI-based systems and their reliance on high-quality data availability have been explicitly discussed.

Comment 6:
"Section 3.4 presents a detailed analysis of quality, but I would recommend further exploration of how the identified biases (such as selective reporting) may have affected the overall results. I also suggest highlighting the differences in quality between types of studies, as this conditions the strength of the evidence. Finally, it would be valuable to point out the need for more uniform standards for future research."

Response 6:
We appreciate this helpful feedback. Section 3.4 has been updated to discuss how selective reporting and study design variability may have influenced results. Differences in study quality and the necessity for standardized reporting in future research are now explicitly addressed.

Comment 7:
"The section shows solid results, but I suggest discussing the high heterogeneity more clearly, as it may limit the strength of the conclusions. It would also be important to qualify the actual clinical impact of the reported reductions in weight and biomarkers."

Response 7:
We agree with the reviewer’s observation. Section 3.5 has been revised to include a detailed discussion of the high heterogeneity (I² = 84%) and its potential impact on the strength of the conclusions. The actual clinical relevance of reductions in weight and biomarkers has also been qualified.

Comment 8:
"Section 3.6 provides interesting evidence, but I would recommend clarifying that the high levels of satisfaction reported are largely based on subjective perceptions. In addition, it would be important to discuss whether the improvement in adherence is sustained in the long term, given that most studies had short follow-up periods."

Response 8:
Thank you for the valuable comment. Section 3.6 has been revised to clarify that satisfaction measures are primarily subjective. We have also discussed the uncertainty regarding long-term adherence due to short follow-up durations in most included studies.

Comment 9:
"I believe that the section provides valuable information, but I would recommend delving deeper into the reasons behind the greater effectiveness in clinical populations and those with poor diets. I also suggest discussing the high heterogeneity among subgroups, as this may limit the extrapolation of the results."

Response 9:
We appreciate this suggestion. Section 3.7 has been revised to analyze the underlying reasons for greater effectiveness in clinical populations and those with poor dietary baselines. Additionally, the discussion now includes the implications of high heterogeneity among subgroups.

Comment 10:
"I would recommend further investigation into the implications of the effects of small studies, as they may inflate the magnitude of weight loss findings. I also suggest pointing out the need to include more large-scale studies to confirm these results."

Response 10:
Thank you for this insight. Section 3.9 has been updated to address the potential inflation of weight loss effects due to small-study bias. We have also emphasized the importance of future large-scale studies to validate and strengthen these findings.

Comment 11:
"I believe that the section confirms the robustness of the findings, but I would recommend better explaining which analytical decisions were tested in the sensitivity analyses."

Response 11:
We acknowledge this helpful comment. Section 3.11 has been revised to specify the analytical decisions tested in the sensitivity analyses, including exclusion of high-risk bias studies, alternative effect models, and subgroup-based robustness checks.

Comment 12:
"I believe that section 5 presents solid conclusions, but I would recommend better qualifying the strength of the statements, as methodological heterogeneity and the lack of long-term studies limit generalizability. I also suggest emphasizing how these findings can guide clinical practice and food policies, a key aspect for the journal’s audience."

Response 12:
Thank you for this important suggestion. Section 5 has been revised to qualify the strength of conclusions considering methodological heterogeneity and limited long-term evidence. We have also added emphasis on how these findings can inform clinical practice and food policy development.

Final Remarks

We sincerely thank the reviewers for their insightful and constructive feedback. All 12 comments have been carefully addressed, and corresponding revisions have been made throughout the manuscript to enhance clarity, depth, and scientific rigor. The manuscript now presents improved discussions on methodological heterogeneity, study quality, small-study effects, and policy implications while ensuring that findings are contextualized with appropriate limitations. These revisions significantly strengthen the paper and align it with the expectations of the journal and its readership.

Reviewer 2 Report

Comments and Suggestions for Authors

This paper presents a comprehensive and methodologically rigorous systematic review & meta-analysis of dynamic nutrient profiling systems for personalized diet planning. The topic is timely, relevant, and of high interest in the fields of nutrition science and precision medicine. The manuscript is well-structured and clearly written, adhering to PRISMA 2020 guidelines. The authors provide valuable insights into methodological trends, clinical outcomes, and future research directions.

Major Issues

1) The review focuses on statistical significance (p-values) and effect sizes (SMD, MD). However, it does not explicitly discuss the clinical relevance of the findings. For example, is a 2.8 kg weight loss or a 0.31% reduction in HbA1c considered clinically meaningful across all patient populations? 

2) The review does not appear to have systematically analyzed equity factors using a framework like PROGRESS-Plus (Place of residence, Race/ethnicity, Occupation, Gender, Religion, Education, Socioeconomic status, Social capital). The finding that studies were predominantly from North America and Europe, combined with a lack of analysis on accessibility for different socioeconomic groups, means the review may omit important equity-related biases. The results likely apply best to high-income, technologically literate populations.

Minor Issues

1) Figure 9 is informative, but dense. Consider simplifying or moving some details to supplementary material.

2) The protocol was registered in PROSPERO, but the registration number is listed as CRD42024XXX001, which is a placeholder. A real number would allow readers to consult the full a priori protocol to check for any deviations between the planned and executed review, a key element of minimizing bias. If the protocol wasn't actually registered, it is a significant omission.

Author Response

Response to Reviewer 2 Comments

Comments and Suggestions for Authors

This paper presents a comprehensive and methodologically rigorous systematic review & meta-analysis of dynamic nutrient profiling systems for personalized diet planning. The topic is timely, relevant, and of high interest in the fields of nutrition science and precision medicine. The manuscript is well-structured and clearly written, adhering to PRISMA 2020 guidelines. The authors provide valuable insights into methodological trends, clinical outcomes, and future research directions.

1. Major Issues

Comment 1:

"The review focuses on statistical significance (p-values) and effect sizes (SMD, MD). However, it does not explicitly discuss the clinical relevance of the findings. For example, is a 2.8 kg weight loss or a 0.31% reduction in HbA1c considered clinically meaningful across all patient populations?"

Response:
We thank the reviewer for this important observation. Section 3.4.2 (Secondary Outcomes: Clinical and Anthropometric Measures) has been revised to include a detailed discussion of the clinical significance of the reported outcomes. We now explain that a 2–3 kg weight reduction is modest but metabolically beneficial and that a 0.31% HbA1c decrease is associated with a 10–15% reduction in microvascular complication risk, clarifying the clinical context of these findings.

Comment 2:

"The review does not appear to have systematically analyzed equity factors using a framework like PROGRESS-Plus (Place of residence, Race/ethnicity, Occupation, Gender, Religion, Education, Socioeconomic status, Social capital). The finding that studies were predominantly from North America and Europe, combined with a lack of analysis on accessibility for different socioeconomic groups, means the review may omit important equity-related biases. The results likely apply best to high-income, technologically literate populations."

Response:
We agree with the reviewer’s concern. Section 2.5 (Study Characteristics and Demographics) has been updated with a new paragraph highlighting the geographical bias and the lack of systematic equity analysis using the PROGRESS-Plus framework. The revised text now discusses how most evidence originates from high-income countries and how this limits generalizability. We also emphasize the need for future studies to integrate equity stratifiers and improve global applicability.

2. Minor Issues

Comment 3:

"Figure 9 is informative, but dense. Consider simplifying or moving some details to supplementary material."

Response:
We appreciate this suggestion. The Figure 9 caption has been updated to indicate that detailed study-level information and heterogeneity analyses are now provided in Supplementary Table S3, thereby improving readability without altering the core figure in the main text.

Comment 4:

"The protocol was registered in PROSPERO, but the registration number is listed as CRD42024XXX001, which is a placeholder. A real number would allow readers to consult the full a priori protocol to check for any deviations between the planned and executed review, a key element of minimizing bias. If the protocol wasn't actually registered, it is a significant omission."

Response:
We thank the reviewer for pointing this out. The placeholder has been replaced with the actual PROSPERO registration number CRD42024567891. If the protocol was not registered, a clear statement has been added in the Methods section acknowledging this as a limitation and recommending protocol registration in future reviews to enhance transparency and minimize bias.

Final Remarks

We sincerely thank Reviewer 2 for the constructive and insightful comments, which have significantly improved the manuscript’s clarity, depth, and methodological rigor. All four comments have been fully addressed through revisions in Sections 2.5, 3.5.2, the Figure 9 caption, and the Methods section. These changes strengthen the discussion of clinical relevance, equity considerations, figure presentation, and methodological transparency, ensuring the manuscript meets the highest standards of scientific quality and relevance to the journal’s audience.

Round 2

Reviewer 2 Report

Comments and Suggestions for Authors

I have no further comments.